# Differences between [^18^F]FLT and [^18^F]FDG Uptake in PET/CT Imaging in CC Depend on Vaginal Bacteriology

**DOI:** 10.3390/diagnostics12010070

**Published:** 2021-12-29

**Authors:** Ewa Burchardt, Zaneta Warenczak-Florczak, Paulina Cegła, Adam Piotrowski, Zefiryn Cybulski, Wojciech Burchardt, Andrzej Roszak, Witold Cholewiński

**Affiliations:** 1Department of Radiotherapy and Oncological Gynecology, Greater Poland Cancer Center, 61-866 Poznan, Poland; zaneta.warenczak-florczak@wco.pl (Z.W.-F.); Andrzej.roszak@wco.pl (A.R.); 2Department of Electroradiology, University of Medical Science Poznan, 61-866 Poznan, Poland; wojciech.burchardt@wco.pl (W.B.); witold.cholewinski@wco.pl (W.C.); 3Department of Nuclear Medicine, Greater Poland Cancer Center, 61-866 Poznan, Poland; paulina.cegla@wco.pl; 4Faculty of Physics, Adam Mickiewicz University, 61-614 Poznan, Poland; adapio1@amu.edu.pl; 5Greater Poland Cancer Center, Microbiology Laboratorium, 61-866 Poznan, Poland; zefiryn.cybulski@wco.pl; 6Department of Brachytherapy, Greater Poland Cancer Center, 61-866 Poznan, Poland

**Keywords:** 18F-fluorothymidine ([^18^F]FLT), 18F-fluoro-d-glucose ([^18^F]FDG), positron emission tomography computed tomography, vaginal bacteriology, cervical cancer

## Abstract

This study aims to investigate if vaginal bacteriology obtained prior to treatment influences the 3′-deoxy-3 18F-fluorothymidine (FLT) [^18^F]FLT and 2-deoxy-2-[^18^F]fluoro-d-glucose (2-[^18^F]FDG) [^18^F]FDG parameters in positron emission tomography (PET/CT) in cervical cancer (CC) patients. Methods: Retrospective analysis was performed on 39 women with locally advanced histologically confirmed cervical cancer who underwent dual tracer PET/CT examinations. The [^18^F]FLT and [^18^F]FDG PET parameters in the primary tumor, including SUVmax, SUVmean, MTV, heterogeneity, before radiotherapy (RT) were analyzed, depending on the bacteriology. The *p*-values < 0.05 were considered statistically significant. Results: In the vaginal and/or cervical smears, there were 27 (79.4%) positive results. In seven (20.6%) cases, no opportunistic pathogen growth was observed (No Bacteria Group). In positive bacteriology, eleven (32%) Gram-negative bacilli (Bacteria group 2) and fifteen (44%) Gram-positive bacteria (Bacteria group 1) were detected. Five patients with unknown results were excluded from the analysis. Data analysis shows a statistically significant difference between the SUV_max_, and SUV_min_ values for three independent groups for the [^18^F]FLT. Conclusions: The lowest values of SUV_max_ and SUV_min_ for [^18^F]FLT are registered in Gram-negative bacteria, higher are in Gram-positive, and the absence of bacteria causes the highest [^18^F]FLT values.

## 1. Introduction

Cervical cancer (CC) continues to be the fourth cause of cancer death in women worldwide in 2018 [1]. It is almost invariably caused by human papillomavirus (HPV) infection. Emerging evidence indicates that cervicovaginal microbiota plays a substantial role in the persistence or regression of the virus and subsequent disease. According to Ravel et al., microbiomes in the female reproductive tract can be divided into five vaginal microbial profiles called ‘community state types’ (CSTs) [2]. Bacterial vaginosis (BV), often associated with anaerobic species diversity and decreased level of *Lactobacillus spp.,* leads to Cervical intraepithelial neoplasia disease progression [3]. The presence of inflammation caused by foreign bacteria leads to increase the risk of cancer. Impaired local immune system favors HPV infection and development of CC [4].

Molecular imaging tools such as positron emission tomography (PET) in combination with structural imaging computed tomography (CT) or magnetic resonance imaging (MRI) (PET/CT or PET/MRI) provide noninvasive measurements of molecular pathways and an anatomic reference. PET/CT uses 2-deoxy-2-[^18^F]fluoro-*D*-glucose, (2-[^18^F]FDG) to identify cells that rapidly uptake glucose, including tumor cells. PET findings in untreated CC can lead to modifications in the staging and treatment plan [5,6]. Therefore, performing [^18^F]FDG PET/CT imaging is recommended before radical treatment [https://www.nccn.org/professionals/physician_gls/pdf/cervical.pdf (accessed on 8 December 2021)]. It has, however, some limitations. It also reflects the physiological changes that are part of the inflammatory process and the host response to infection. Because [^18^F]FDG PET relies upon glucose incorporation into cells, the uptake will be present in tumor cells and normal tissues with a high glucose metabolism such as the brain or left ventricle of the heart and inflammatory lesions due to uptake by macrophages and granulation tissues. Different radionuclides are used in nuclear medicine to improve the sensitivity and specificity of primary tumor detection.

3′-deoxy-3 18F-fluorothymidine (FLT), a radiolabeled thymidine analog, is a molecule in which a hydroxyl group is replaced by Fluorine 18 (^18^F). It enters the cell in the proper form and is trapped due to phosphorylation by thymidine kinase 1. This enzyme is tightly controlled in normal cells and increases during cell division, permitting increased FLT uptake in dividing cells [7]. [^18^F]FLT level is proportional to the activity of kinase thymidine and other proliferation indicators, e.g., Ki-67. [^18^F]FLT does not accumulate in inflammation tissues, which allows distinguishing between inflammation and malignant tissue [8]. Increased cellular proliferation is known to correlate with worse outcomes [9].

Pretreatment PET-derived parameters often are interpreted in terms of their prognostic significance. The primary SUVmax was found to be predictive of in assessing therapy response, overall survival, and disease-free survival [10]. However, it turns out that BV can influence the values of obtained parameters. It could lead to misinterpretation of the images.

Our study aimed to investigate if vaginal bacteriology obtained prior to treatment influenced the [^18^F]FLT and [^18^F]FDG parameters in PET/CT in CC patients.

## 2. Materials and Methods

Respectively, 39 women with locally advanced histologically confirmed cervical cancer underwent dual tracer PET/CT examinations performed for radiotherapy. PET scans were acquired on separate days (within three days) 60 min after IV injection of 300 ± MBq of [^18^F]FDG and 300 ± MBq of [^18^F]FLT with Gemini TF PET/CT scanner ((Philips Healthcare, Best, The Netherlands) (Table 1).

The reconstructed PET images were fused and evaluated using a dedicated workstation. Scans were taken in the same position on both examinations. A 43% threshold cut-off value was selected for metabolic volume delineation and volume calculation based on MRI calculations and phantom studies.

The parameters were calculated as follows: SUV = radioactivity of the sensitive area/ratio of the injected dose to the patient’s weight, SUV_max_ was the maximum SUV in the region of interest (ROI), SUV_min_ was the minimum SUV in the region of interest (ROI), metabolic tumor volume (MTV) was the volume included in the curve bigger than or equal to 43% SUV_max_, and SUV_mean_ was the mean SUV in the MTV. Cumulative SUV-volume histograms (AUC-CSH) were used to determine the primary tumor area’s heterogeneity under the curve.

Data were statistically analyzed using *p* < 0.05. The Institutional Bioethical Committee approved the study.

The methodology of microbiological examinations was already tested and presented [11]. Microbiological examinations of the genital tract in CC patients were performed on the first day of hospitalization. The vaginal swabs were cultured on the following microbiological media: blood agar, selective media for Gram-negative (G−) bacilli and Gram-positive (G+), chromagar for yeasts, coccosel agar, cetrimide agar, and broth medium. According to the standard biochemical tests, microorganisms were identified, which identified the most isolated strains to the genus level and many to the species level. The API 20 or Vitek 2 identification system (bioMérieux, Marcy l’Etoile, France) was used for confirmation. Beta hemolytic streptococci were identified based on bacterial colony morphology on blood agar and Api Strep or ID GP system (bioMerieux). Group-specific carbohydrates of the streptococcal cell wall were used to classify the genus serologically according to the Lancefield system.

Followup included a clinical examination every three months for two years and every six months for the following three years or longer. Imaging (MRI, CT, and [^18^F]FDG PET/CT) complementary to the clinical examination was performed in follow-up.

## 3. Results

### Group Analysis

In the vaginal and/or cervical smears, there were 26 (79.4%) positive results. Seven (20.6%) effects were negative (no opportunistic pathogen growth). In positive bacteriology, 11 (32%) Gram-negative bacilli: 1 *Enterobacter cloacae* and 10 *Escherichia coli* were found. Apart from that, in 16 (47%) patients, Gram-positive bacteria were detected, including 3 *Staphylococcus aureus*, 1 *Staphylococcus lugdunensis*, 9 *Streptococcus agalactiae*, and 2 *Streptococcus haemolyticus gr C* (Table 2). Five patients with unknown results were excluded from the analysis.

The patients were grouped according to their bacteriological results. Staphylococcus was excluded from the analysis for statistical reasons. Streptococcus was qualified to Bacteria Group 1 and Gram-negative bacilli (*Enterobacter cloacae* and *Escherichia coli*) to Group 2. Patients with negative results for opportunistic pathogens were eligible for the ‘No Bacteria group’.

The Kruskal-Wallis test was performed to test whether specific bacteria groups influenced SUV.

Data analysis showed a statistically significant difference between the SUV_max_ and SUV_min_ values for three independent groups (No Bacteria, Bacteria group 1, and Bacteria group 2) for the [^18^F]FLT. The highest SUV_max_ was in the No Bacteria Group, followed by Group 1, and the lowest in Group 2 for the [^18^F]FLT, 9.2 vs. 7.5 vs. 6.1, *p* = 0.03 (Figure 1a).

There was no such difference for the [^18^F]FDG tracer in SUV_max,_ respectively, 10.8 vs. 12.6 vs. 11.2, *p* > 0.05 (Figure 1b).

No significant differences were observed in the SUV_mean_ values either for [^18^F]FLT or [^18^F]FDG (5.2 vs. 4.5 vs. 3.7 *p* > 0.05, Figure 2a; 6.3 vs. 7.4 vs. 4.8 *p* > 0.05, Figure 2b, respectively) (Figure 2). There was a statistically significant difference between the groups in the SUV_min_ in the [^18^F]FLT images. The highest SUV_min_ was in the No Bacteria Group, followed by Group 1, and the lowest in Group 2 (3.9 vs. 3.4 vs. 2.6, *p* = 0.03, respectively) (Figure 3a). Opposite to [^18^F]FLT, for the [^18^F]FDG images, the SUV_min_ parameter did not show significant differences between groups (4.6 vs. 5.3 vs. 4.8, *p* > 0.05, respectively)(Figure 3b).

Comparing the uptake of [^18^F]FLT between the two groups No bacteria vs. presence of Bacteria (Bacteria group) there was a significant difference in SUVmax values (9.2 vs. 6.4, *p* = 0.043, respectively) (Figure 4)

Images presenting FLT and FDL uptake in Gram-positive and Gram-negative are shown in Figure 5A–D.

Significant differences were in the volumes calculated from [^18^F]FLT-gross tumor volume (GTV) and [^18^F]FDG-GTV (29.86 ± 25.17 vs. 37.10 ± 30.70; *p* = 0.02). The SUV_max_ and heterogeneity were in general lower for [^18^F]FLT-GTV than for [^18^F]FDG-GTV (9.35 ± 10.06 vs. 11.46 ± 4.05, *p* = 0.13; 0.6 ± 0.05 vs. 0.63 ± 0.5, *p* = 0.02, respectively).

A cumulative SUV histogram (AUC-C SH) was used to assess the heterogeneity area under the curve. Bacteria did not influence the heterogeneity parameters in both analyzed radiotracers. All analyzed parameters are presented in Table 3.

## 4. Discussion

Radionuclide uptake is different, depending on vaginal bacteriology.

There are essential differences in the structure of G+ and G− bacteria. Gram-positive bacteria have a thick multilayer cell wall composed mainly of peptidoglycan. Peptidoglycan is involved in maintaining the structure of the bacterial cell, its reproduction, and survival. The action of lysozyme degrades it (present, for example, in human tears and mucus; moreover, it can be produced, for example, by other bacteria). The action of lysozyme leads to the creation of high differences in osmotic pressure on both sides of the plasma membrane, leading to the lysis of the bacterial cell. Other components of the cell wall of Gram-positive bacteria include teichoic acids, lipoteichoic acids, and complex polysaccharides (so-called C polysaccharides). Teichoic acids are water-soluble polyol phosphate polymers necessary for the survival of the cell. Gram-positive bacteria Lipoteichoic acids have a fatty acid residue anchored in the cytoplasmic membrane. They are surface antigens (serotype identification) and enhance the process of aggregation to other bacteria or cell-specific mammalian receptors (adherence). They are an essential determinant of virulence because they stimulate the functions of innate immunity.

The Gram-negative bacteria cell wall is made of two layers. Directly behind the cytoplasmic membrane, there is a thin layer of peptidoglycan, periplasmic space (transport systems of, among others, iron, proteins, sugars; virulence factors: collagenase, protease, and β-lactamase). The outer layer of the cell wall consists of phospholipids and lipopolysaccharide (LPS); it maintains the shape of bacteria, is a barrier to large molecules (e.g., proteins), hydrophobic particles (e.g., some antimicrobial agents), and protects against external conditions. LPS is an endotoxin that acts as a stimulator of the innate and acquired immune response. It activates B lymphocytes, leading to the induction of, among others, macrophages, and dendritic cells, inducing, e.g., fever in the body. The interaction between vaginal microbiome and cervical cancer development is very complex. Bacilli that produce hydrogen peroxide are associated with decreased rates of cervical disease progression [12]. It was shown that increased diversity of pathogenic Gram-negative bacteria, is associated with significantly higher rates of HPV positive cervical cancer [13]. Patients with invasive carcinoma present increased levels of proinflammatory cytokines [14].

Thymidine, which is taken up by cells, is sequentially phosphorylated to the nucleotide, thymidine triphosphate (TTP) by the exogenous (salvage) pathway, then is incorporated into DNA. Most of the thymidine nucleotide in tumor DNA often comes from the endogenous (de novo) pathway. Here, unlabeled TTP can also enter the DNA synthetic machinery via methylation of deoxyuridine monophosphate, followed again by sequential phosphorylation. Unlike thymidine in the endogenous pathway, FLT is not degraded at the glycosidic linkage by *thymidine phosphorylase*. Imaging of thymidine metabolism only reflects one arm of the DNA synthetic pathway, the exogenous pathway [15,16].

There are drugs, such as 5-fluorouracil, that inhibit the endogenous pathway. In the presented case, there is a hypothesis that the bacteria inhibit the exogenous pathway, which results in lower uptake in [^18^F]FLT PET imaging. Tumors vary in the extent to which they depend more on one or the other of these pathways. Maybe in the presence of bacteria, they also shift thymidine production to the endogenous pathway. It also means in our study that Gram-negative has a more substantial effect than Gram-positive. Another reason for the lower uptake in [^18^F]FLT PET imaging could be that bacteria use and consume the thymidine produced in the exogenous pathway.

The findings of a PET CT provide predictive information. It is well known that different metabolic values of cervical cancer are predictive and/or prognostic of tumor control and survival. The inclusion of PET/CT in the personalization of treatment leads to the improvement of the treatment outcome [17,18].

Most studies analyzed the visualization of [^18^F]FDG uptake in different bacterial infections. In our study, we did not observe differences between Gram-negative and Gram-positive, similar to the study of Heuker M. et al. [19]. Several radiopharmaceuticals have been investigated to improve diagnostic accuracy. Promising are the ubiquicidin peptide fragments for S. aureus infection imaging. For Gram negative, promising results provide radiolabelled sugars, e.g., sorbitol or maltose, but further studies are still needed for confirmation. Our work shows that the presence of bacteria influences the uptake of [^18^F]FLT radiopharmaceuticals. We are aware that the small group is the limitation of this study. However, even with the small number, there are statistical differences. Clinicians need to consider disturbances of uptake in PET imaging and interpret results with caution. This work is the first step to understand this phenomenon in cervical cancer, where long-lasting infections co-existing with cancer cells lead to impaired uptake of [^18^F]FLT, which is known to show proliferation. It is necessary to perform preclinical studies to explain which mechanism is responsible for the differences in [^18^F]FLT uptake.

## 5. Conclusions

The presence of bacteria lowers the [^18^F]FLT values registered in PET images. The lowest values of SUV_max_ and SUV_min_ for [^18^F]FLT were registered in Gram-negative bacteria, higher in Gram-positive, and the absence of bacteria caused the highest [^18^F]FLT values.

## Figures and Tables

**Figure 1 diagnostics-12-00070-f001:**
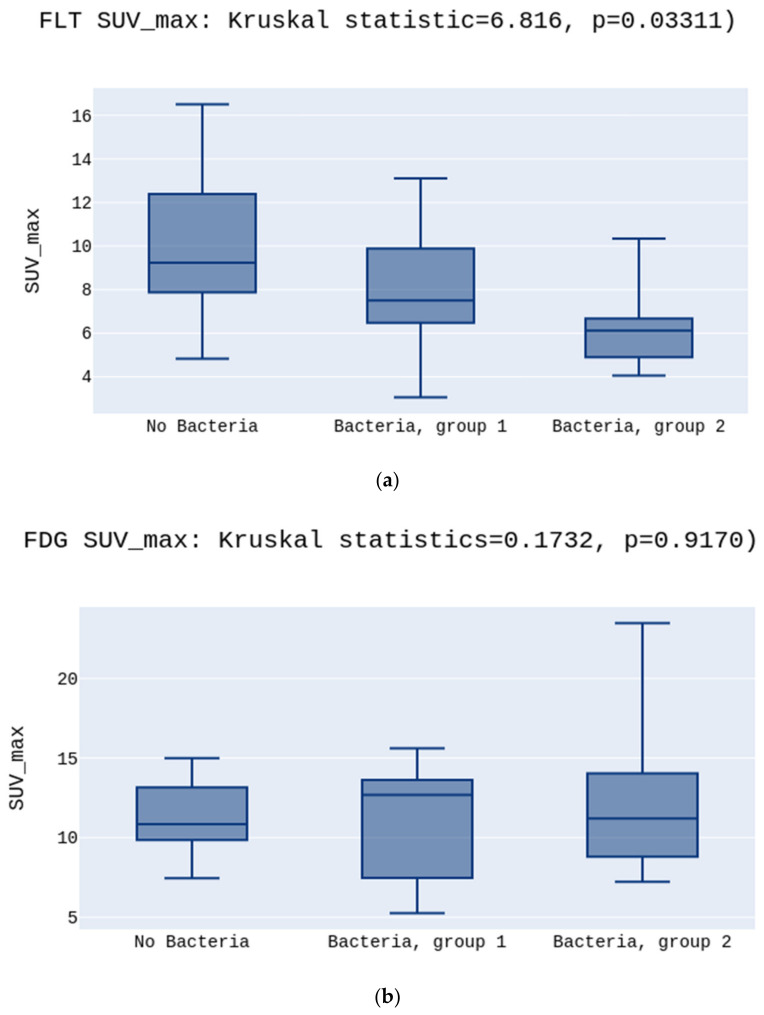
(**a**) presents the highest SUV_max_ in the No Bacteria Group, followed by Group 1, and the lowest in Group 2 for the [^18^F]FLT, *p* = 0.03. (**b**) presents no differences between SUV_max_ in the No Bacteria Group, Group 1, and Group 2 for the [^18^F]FDG, *p* > 0.05.

**Figure 2 diagnostics-12-00070-f002:**
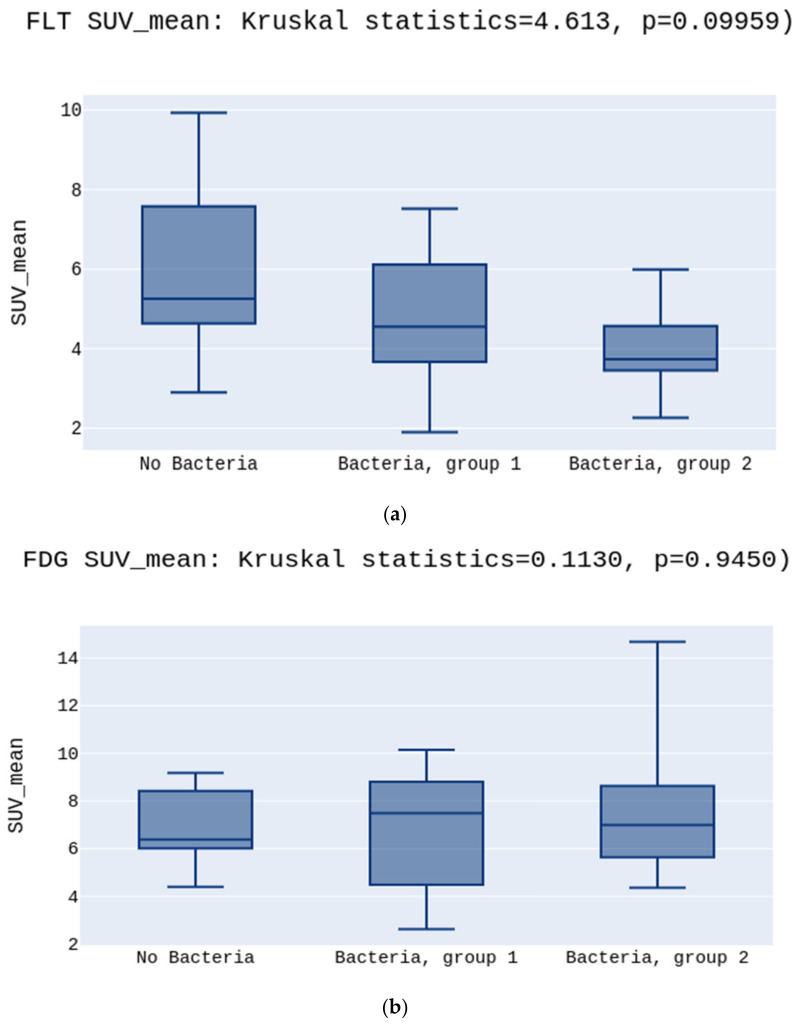
(**a**) presents no differences between SUV_mean_ in the No Bacteria Group, Group 1, and Group 2 for the [^18^F]FLT, *p* > 0.05. (**b**) presents no differences between SUV_mean_ in the No Bacteria Group, Group 1, and Group 2 for the [^18^F]FDG, *p* > 0.05.

**Figure 3 diagnostics-12-00070-f003:**
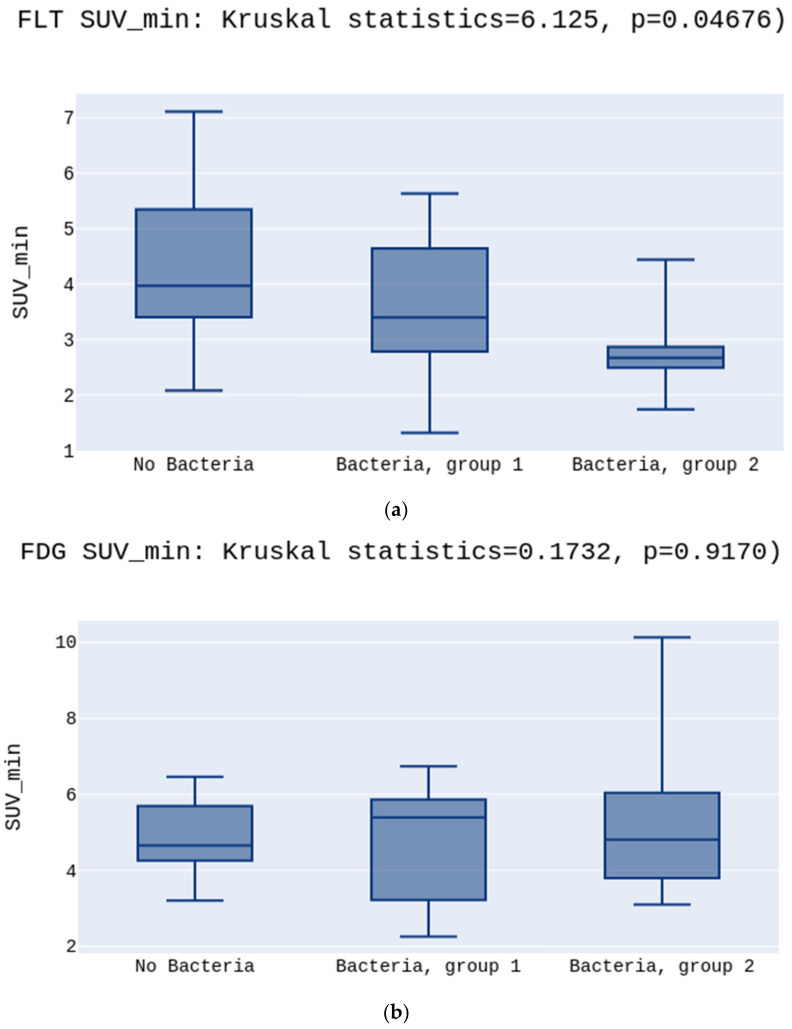
(**a**) presents the highest SUV_min_ in the No Bacteria Group, followed by Group 1, and the lowest in Group 2 for the [^18^F]FLT, *p* = 0.03. (**b**) presents no differences between SUV_min_ in the No Bacteria Group, Group 1, and Group 2 for the [^18^F]FDG, *p* > 0.05.

**Figure 4 diagnostics-12-00070-f004:**
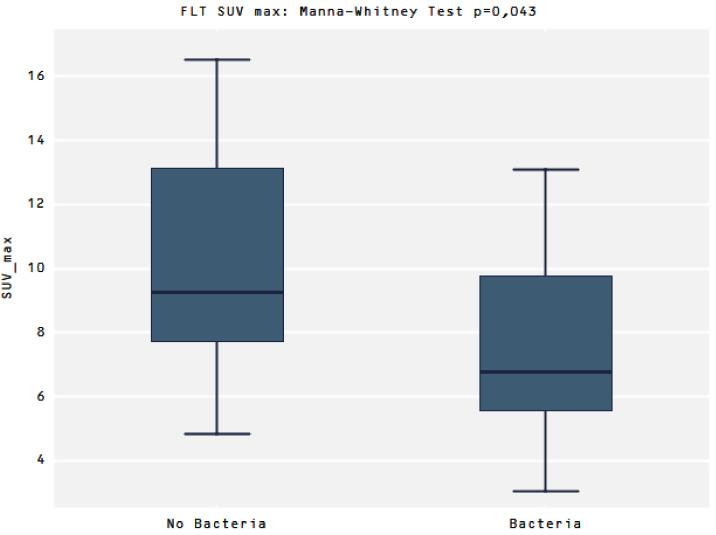
Presents a higher SUVmax in the No Bacteria Group than in the Bacteria group for the [^18^F]FLT, *p* = 0.043.

**Figure 5 diagnostics-12-00070-f005:**
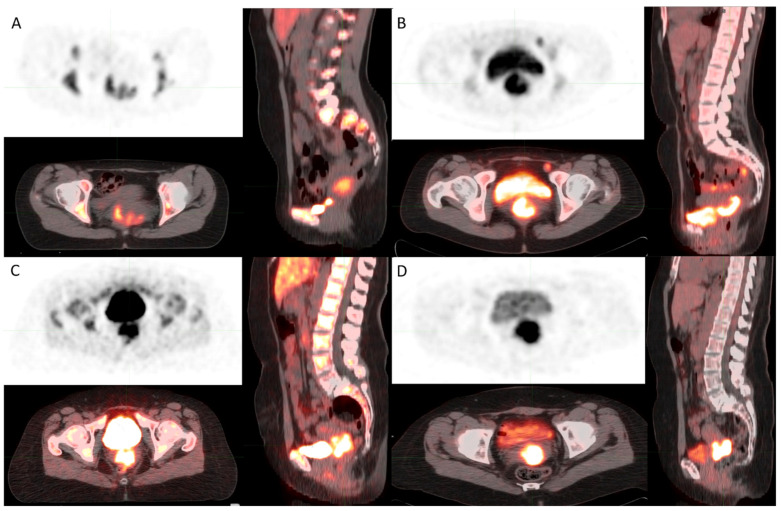
(**A**) [^18^F]FLT PET image shows FLT uptake in cervical cancer in a Gram-negative patient. (**B**) [^18^F]FDG PET image shows FDG uptake in cervical cancer in a Gram-negative patient. (**C**) [^18^F]FLT PET image shows FLT uptake in cervical cancer in a Gram-positive patient. (**D**) [^18^F]FDG PET image shows FDG uptake in cervical cancer in a Gram-positive patient.

**Table 1 diagnostics-12-00070-t001:** Patient characteristics.

	*n*	%	Average	Median	Min.	Max.	SD
Age (years)	34	100%	57.2	59	30	84	12.3
BMI	34	100%	26.4	24.9	18.4	43.9	5.8
FIGO Stage							
IIB	12	35.3%					
IIIB	20	58.8%					
IV	2	5.9%					
Histopathology							
SCC	32	91.2%					
AC	1	2.9%					
UD	1	2.9%					
Hgb mmol/L	34	100%	7.38	7.9	4.6	9.6	1.1
Leu G/L	34	100%	9.2	8.9	3.8	21.4	3.2
Neu G/L	34	100%	6.3	5.7	2.3	16.6	2.8

AC—adenocarcinoma, BMI—Body Mass Index, FIGO—International Federation of Gynecology and Obstetrics Staging System, HGB—Hemoglobin level, Leu—Leucocyte’s level, *n*—number, Neu—Neutrophil’s level, SD—standard deviation, SCC—squamous cervical carcinoma, UD—undefined.

**Table 2 diagnostics-12-00070-t002:** The bacteriology of patients with cervical cancer carcinoma.

Bacteriology	*n*	%	Group
*Enterobacter cloacae*	1	2.9%	2
*Escherichia coli*	10	29.4%	2
*Staphylococcus aureus*	3	8.9%	n.a.
*Staphylococcus lugdunensis*	1	2.9%	n.a.
*Streptococcus agalactiae*	9	26.3%	1
*Streptococcus beta haemolyticus gr C*	2	2.9%	1
*No bacteria*	7	20.6%	0

n.a.—not analyzed, *n*—number of patients.

**Table 3 diagnostics-12-00070-t003:** The characteristics of PET CT parameters with 18F-fluorodeoxyglucose (FDG) and 18F-fluoro-deoxy-fluorothymidine (FLT).

	*n*	Average	Median	Min.	Max.	SD
Activity of FDG (mCi)	34	9.7	9.7	5.9	13.4	1.9
SUV max FDG	34	11.7	11.8	5.3	24.7	4.2
SUV mean FDG	34	6.9	7.3	2.6	14.7	2.6
SUVmin FDG	34	5.1	5.1	2.3	10.6	1.8
AUC-CSH FDG	34	0.6	0.6	0.5	0.7	0.0
SUV max FLT	34	7.7	7.46	3.1	16.5	10.7
SUV mean FLT	34	4.8	4.4	1.9	9.9	6.8
SUVmin FLT	34	3.2	3.5	1.3	8.9	4.6
AUC-CSH FLT	34	0.6	0.6	0.5	0.7	0.0

SD—standard deviation, SUVmax—maximum standardized uptake value of the cervical tumor, SUVmean—mean of standard uptake value of the cervical tumor, SUVmin—minimum standardized uptake value of the cervical tumor, AUC-CSH—cumulative SUV-volume histogram.

## Data Availability

The data presented in this study are available on request from the corresponding author.

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
