# Peer review of "Differences between [18F]FLT and [18F]FDG Uptake in PET/CT Imaging in CC Depend on Vaginal Bacteriology"

_diagnostics, 2021, doi:10.3390/diagnostics12010070_

Round 1
Reviewer 1 Report
After modficiations, the paper looks fine. There is only one small correction to make: line 41, please write in English
Reviewer 2 Report
I think that the manuscript improved significantly.
Reviewer 3 Report
Ewa Burchardt et al. in ” Differences between [18F] FLT and [18F]FDG uptake in PET/CT 2 imaging in CC depend on vaginal bacteriology” investigated the role of vaginal bacteriology prior to treatment to understand its involvement on 3'-deoxy-3 18F-fluorothymidine (FLT) [18F] FLT and 2-deoxy-2- [18F] fluoro-D-glucose ( 2- [18F] FDG) [18F] FDG parameters, in positron emission tomography (PET / CT) in patients with cervical cancer (CC). Absorption of radionuclides can be influenced by vaginal bacteriology, which therefore plays an important role in this context. According to the authors, the absorption of [18F] FLT radiopharmaceuticals is influenced by the presence of bacteria. For this reason, physicians should consider absorption disturbances in PET imaging for accurate interpretation of results. This study highlights the importance of vaginal bacteriology in cervical cancer, where infections can be present simultaneously with cancer cells, leading to a reduction in the absorption of [18F] FLT. The lowest values ​​of SUVmax and SUVmin for [18F] FLT are recorded in Gram-negative bacteria, while the highest they are in Gram-positive and the absence of bacteria produces the highest values ​​of [18F] FLT.
The paper is well written and very original.
This manuscript is a resubmission of an earlier submission. The following is a list of the peer review reports and author responses from that submission.
Round 1
Reviewer 1 Report
The work presented in the manuscript by Burchardt et al is, in its essence, interesting. The authors sought to explore the relationship between cervical cancer and the presence of bacteria in the vaginal tract. However, the results are very poorly presented, as well as the discussion. The manuscript needs a lot of improvement before it can be accepted for publication. It needs editing (for instance the names of the bacteria should be in italic) and English proofreading. There are other problems like the isolation of bacteria. Lactobacilli are mentioned in the introduction, but the authors do not mention it in the results? Maybe because of the conditions used for isolating bacteria? Then why mention it at all in the introduction? It would be better to focus on the role of the bacteria that was indeed isolated in cervival cancer and/or the vaginal tract. Revise bacterial taxonomy - Streptococcus haemolyticus might be Streptococcus pyogenes? In the words of the authors: "Our work shows that the presence of bacteria influences the uptake of 221 [18F]FLT radiopharmaceuticals. Clinicians need to consider disturbances of uptake in PET 222 imaging and interpret results with caution. This work is the first step to understand this 223 phenomenon in cervical cancer, where long-lasting infections co-existing with cancer cells 224 lead to impaired uptake of [18F]FLT, which is known to show proliferation. It is necessary 225 to perform preclinical studies to explain which mechanism is responsible for the differ-226 ences in [18F]FLT uptake". These conclusions are interesting and novel to the field, but the manuscript does not lead the reader to this conclusion.
Reviewer 2 Report
In the present manuscript, “Differences between [18F] FLT and [18F] FDG uptake in PET/CT imaging in CC depend on vaginal bacteriology,” the authors investigated the relationship between vaginal flora and PET imaging. However, there are several concerns, and detailed comments are listed below.
- The clinical significance of this study is unclear. Please mention it in the Introduction section. Do low SUVs due to the bacterial flora bring clinical disadvantages?
- Based on the FIGO staging of cervical cancer, the vaginal environment of stage III is considered to vary greatly depending on the subclassification, and stage IIIA disease involves the lower third of the vagina. Did the authors perform any analysis on stage IIIA?
- In lines 117 and 118, the authors described that “Staphylococcus were excluded from the analysis due to statistical reasons.” What is the statistical reason?
- The authors should plot the values ​​for each case on box plots shown in Figure 3. The box plots appear to have no remarkable differences between the groups. Moreover, statistical errors may occur due to individual differences. How about the comparison between the no bacteria and bacteria groups?
Reviewer 3 Report
Ewa Burchardt et al. in ” Differences between [18F] FLT and [18F]FDG uptake in PET/CT 2 imaging in CC depend on vaginal bacteriology.”, evaluated the role of vaginal bacteriology prior to treatment to understand its involvement on 3'-deoxy-3 18F-fluorothymidine (FLT) [18F] FLT and 2-deoxy-2- [18F] fluoro-D-glucose ( 2- [18F] FDG) [18F] FDG parameters, in positron emission tomography (PET / CT) in patients with cervical cancer (CC). Vaginal bacteriology, is important because can influence the uptake of radionuclides. Indeed, the authors demonstrate that the presence of bacteria influences the absorption of [18F] FLT radiopharmaceuticals. Therefore, clinicians should consider uptake disturbances in PET imaging for accurate interpretation of results. This work highlights the importance of vaginal bacteriology in cervical cancer, where infections can coexist with cancer cells leading to reduced [18F] FLT uptake. The lowest values of SUVmax and SUVmin for [18F]FLT are registered in Gram-negative bacteria, while higher are in Gram-positive, and the absence of bacteria produces the highest [18F]FLT values.
The paper is well written and original, but Authors should add:
- (1) before This (line 16).
- (2) before Methods (line 19).
Authors should remove the point after Fig. 1a in the text, (line 146).
